

# Developmental instability in wild Nigerian olive baboons (*Papio anubis*)

Kara C. Hoover[1,2], Emily Gelipter[3], Volker Sommer[4,5] and Kris Kovarovic[3]

[1] Department of Anthropology, University of Alaska Fairbanks, Fairbanks, AK, United States of America
[2] Department of Biochemistry and Molecular Biology, University of Alaska—Fairbanks, Fairbanks, AK, United States of America
[3] Department of Anthropology, Durham University, Durham, United Kingdom
[4] Gashaka Primate Project, Serti, Taraba, Nigeria
[5] Department of Anthropology, University College London, University of London, London, United Kingdom

Corresponding author
Kara C. Hoover,
kchoover@alaska.edu

## ABSTRACT

**Background:** Developmental instability in archaeological samples can be detected through analysis of skeletal and dental remains. During life, disruptions to biological internal homeostasis that occur during growth and development redirect bodily resources to returning to homeostasis and away from normal processes such as symmetrical development. Because dental enamel does not remodel in life, any deviations from normal development are left behind. Even subtle disturbances to developmental trajectory may be detected in asymmetrical development of traits, specifically a random variation in sides termed fluctuating asymmetry. Human dental fluctuating asymmetry studies are common, but here we investigate the permanent dentition of a non-human primate *Papio anubis*, for potential fluctuating asymmetry relative to sex, weaning, and reproductive maturity. The sample stems from an outlier population that lives in the wettest and most humid habitat of any studied baboon group.

**Methods:** The skulls of adult baboons were collected after their natural death in Gashaka Gumti National Park, Nigeria. The permanent dentition of antimeric teeth (paired) were measured for maximum length and breadth using standard methods. The metrics were analyzed to assess the presence of fluctuating asymmetry in adult permanent mandibular and maxillary dentition. Measurement error and other forms of asymmetry (antisymmetry, directional asymmetry) were considered and dental measures expressing true fluctuating asymmetry were used to address three research questions.

**Results:** Males exhibit greater fluctuating asymmetry than females, suggesting that males experience greater overall instability during the developmental period. While weaning is not more stressful than other life history stages for males and females (using the first molar fluctuating asymmetry index as a proxy compared to other teeth), it is more stressful for females than males. The onset of reproduction is also not more stressful than other life history stages for males and females (using the third molar fluctuating asymmetry index as a proxy compared to other teeth), but it is more stressful for males than females. We explore possible explanations for these findings in the discussion.

# INTRODUCTION

*Papio* (baboons), one of the most intensively studied primate taxa, inhabit a wide range of habitats across Africa. Most troops inhabit dry, open savanna in East and South Africa, while habitats that include forests are rare. Our study focuses on a sample of olive baboons in Gashaka Gumti National Park in northeastern Nigeria. These monkeys are outliers in terms of geography, climate and local environment, inhabiting the wettest and most humid habitat of any studied group (*Sommer & Ross, 2011*). Compared to savanna-dwelling baboons, their troop sizes are significantly smaller (*Higham et al., 2009*), possibly because forest fruits occur in clumped patches that may be difficult for larger groups to exploit. Additionally, many troops across the park raid maize and other agricultural crops to varying extents (*Warren et al., 2011*). These combined factors have implications for reproduction, life history and survival. For example, inter-birth intervals are longer here than elsewhere (*Ross et al., 2011*), and crop-raiding troops have both higher energy intake and reproductive success rates (*Lodge et al., 2013*).

A study of fecal glucocorticoid in Gashaka Gumti female baboons suggests they experience unusual amounts of thermoregulatory and nutritional stress throughout their lives, and this stress varies seasonally relative to rainfall and food availability (*MacLarnon et al., 2015*). No data are yet available on stress in male baboons. Permanent teeth develop *in utero* and record interruptions to growth and development during periods of developmental instability, which may leave permanent marks in the teeth such as fluctuating asymmetry (FA) (*Van Valen, 1962*), dental enamel defects (*Sarnat & Schour, 1942, 1941*), and variation in molar cusp morphology (*Corruccini & Potter, 1981*; *Riga, Belcastro & Moggi-Cecchi, 2014*). There is no correlation between dental enamel defects and FA in either archaeological samples (*Hoover et al., 2005*) or twin studies (*Corruccini, Townsend & Schwerdt, 2005*). Enamel defects may be better at recording nutritional stress due to their association with famine (*Zhou & Corruccini, 1998*) and weaning (*Katzenberg, Herring & Saunders, 1996*; *Ungar, Crittenden & Rose, 2017*), but FA is an established proxy for and a broader signal of developmental instability, which may arise from external factors (*e.g.*, climate change, resource shifts) and intrinsic ones (*e.g.*, genetic buffering) (see *Markow, 1994*; *Polak, 2003*; *Frederick & Gallup, 2007*; *Hoover & Hudson, 2016*; *Leamy & Klingenberg, 2005*).

FA is manifest in paired traits as non-directional deviations from perfect symmetry with equal mean development on both sides (*Van Valen, 1962*). The underlying biological theory behind FA as a proxy for developmental instability is allied with concepts of homeostasis and canalization. During growth and development, traits may canalize (reach their final form) in an internal environment where disruptive stochastic processes (developmental noise) and the capacity to resist disruption (developmental stability or homeostasis) become unbalanced (*Waddington, 1942*). The result is non-directional bilateral variation of traits (*Van Valen, 1962*). These minor deviations from normal growth can be measured by variance across the midline of traits (*Van Valen, 1962*). Thus, presence

and degree of FA reflect an individual's ability to canalize a trait despite genetic and/or environmental stresses that disrupt the normal phenotypic trajectory.

Because primates have longer developmental stages compared to most other mammals in which they can record periods of instability in the hard tissues of the body (*Gingerich & Schoeninger, 1979*) and the development of molars overlaps in time with the life history events of interest (weaning, reproductive maturity) (*Fortman, Hewett & Bennett, 2002*; *Hlusko & Mahaney, 2009*; *Phillips-Conroy & Jolly, 1988*), we have an opportunity to identify if there are peaks of instability associated with specific developmental ages. Weaning, for example, occurs during the developmental window of the first molar and is associated with increased dental stress markers in both human and non-human primates (*Kelley & Schwartz, 2010*; *Smith et al., 2013*). In the case of weaning (and presumably other stresses), differences exist between captive groups and those in the wild where developmental delay is common due to greater variance in environmental stress (*Zihlman, Bolter & Boesch, 2004*). In nonhuman primates, most published data are neither collected on dentition nor used to examine developmental differences based on sex or life history (*Atkinson, Rogers & Cheverud, 2016*; *Boulton & Ross, 2013*; *Hallgrimsson, 1993*; *Leigh & Cheverud, 1991*; *Little et al., 2012*; *Newell-Morris, Fahrenbruch & Sackett, 1989*; *Reeves, Auerbach & Sylvester, 2016*; *Sefcek & King, 2007*; *Waitt & Little, 2006*; *Willmore, Klingenberg & Hallgrímsson, 2005*).

A few studies examining FA in the dentition of nonhuman primates have suggested that traits under sexual selection exhibit greater asymmetry, meaning that they are more susceptible to developmental stress (*Manning & Chamberlain, 1993*; *Manning & Chamberlain, 1994*). Human studies, however, have not indicated a clear signal of sex-based differences in FA (*e.g.*, *Garn, Lewis & Kerewsky, 1965*; *Harris & Nweeia, 1980*; *Kieser, Groeneveld & Preston, 1986*; *Perzigian, 1981*). Because humans exhibit greater FA than other apes (*Frederick & Gallup, 2007*), we need a wider comparative dataset on nonhuman primate FA to understand the range of its manifestation across species and how it explains sex differences in developmental instability due to life history, habitat and social system variation. If we can identify a primate-wide pattern to FA (starting with this study and continuing with others), we will then know to either build hypotheses based on primate-wide trends in FA (in the absence of data on a particular species, for example) or to build hypotheses relative to a species of interest. We contribute the first data on dental FA in baboons and answer the following life history questions:

1. Is there a difference in developmental instability based on sex? Gashaka Gumti baboons live at the edge of their species' geographic and ecological range (*Sommer & Ross, 2011*) and high levels of stress have been reported in adult and subadult females, as measured by fecal glucocorticoid (*MacLarnon et al., 2015*). There are no comparative data available for males and the limited data in nonhuman primate dental FA suggest that traits under sexual selection in both males and females are more vulnerable to FA (*Manning & Chamberlain, 1993*; *Manning & Chamberlain, 1994*). So we do not have an expectation.

   a. Hypothesis: There are no differences in FA between the sexes.

2. Is weaning a stressful time compared to other developmental stages? In baboons, suckling ceases around 15 months. This is prior to the eruption of the first molar, which is estimated to occur around 19.5 months (*Dirks & Bowman, 2007*). The first molar will be diagnostic in answering this question. The null hypothesis is that there is no difference between the FA index for first molars and the FA index for all remaining teeth. Because weaning stress has been previously reported in baboons (*Dirks et al., 2002*; *Rhine et al., 1985*), we expect that first molar indices will have significantly higher FA index than any of the remaining variables.

   b. Hypothesis: FA values for first molars are higher than FA values for other teeth.

3. Are there sex-based differences in reproductive stress? In baboons, menarche begins around 4.3 years and first reproduction at 6.1 years, both of which are prior to the eruption of the third molar at approximately 6.8 years of age (*Dirks & Bowman, 2007*). This suggests the third molars will be diagnostic in answering this question. Given that sexually selected traits may be more susceptible to environmental stress (*Manning & Chamberlain, 1994*) and because reproductive stress is generally high in females due to extra demand on resources, we expect female third molars will exhibit greater FA than male third molars.

   c. Hypothesis: FA values for female third molars are higher than those for males.

## MATERIALS

We analyzed data collected from olive baboons (*Papio anubis*) inhabiting Gashaka Gumti National Park (06°55′–08°13′ N and 011°13′–012°11′ E) in northeastern Nigeria. Permits for research at Gashaka Gumti National Park were awarded (VS) by the Nigeria National Park Service (NPH/GEN/378/V/504). The reserve extends over 6,731 km$^2$ and represents the northern edge of the Gulf of Guinea forests and the Cameroonian Highlands, with the highest peak rising to 2,416 m (*Sommer & Ross, 2011*). The park is surrounded by villages that practice subsistence farming and includes various enclaves inhabited mainly by settled Fulani cattle herders. GGNP baboons live close to the southern edge of the western biogeographical distribution of the species (*Zinner et al., 2011*). Baboons are replaced by large forest-dwelling monkeys (mainly drills), a short distance from the park to the south.

In GGNP, pronounced annual wet and dry seasons correspond with fluctuations in temperature and humidity (*Sommer & Ross, 2011*). Based on weather station data for two study sites collected from 2000–2014 (Kwano at 583 m above sea level and Gamgam at 320 m above sea level), mean minimum temperature is 20.9 °C and mean maximum 32.5 °C. Five months with very little or no rainfall are followed by heavy downpours from mid-April to mid-November that constitute 96.3% of all precipitation (annual mean 1,945 mm, range 1,681–2,337 mm). Among baboon field sites, GGNP is an extreme outlier with regards to rainfall, representing the wettest of all baboon study sites to date (*Higham et al., 2009*).

Our study sample is from baboon skulls that were delivered to the Gashaka Primate Project research station at Kwano by locals and park rangers between 2008 and 2013. The majority were found opportunistically while working in fields or during ranger patrols. Skulls were not accepted if there was evidence (*e.g.*, bullet holes) that the animals had been killed by humans (which is illegal inside the park). Thus, our sample represents a natural death sample. Informal assessments of weathering on bone in the field indicated bones were at weathering stages 0–2, which suggests the material was deposited over approximately 6 years (*Behrensmeyer, 1978*). The skulls originated from six localities within GGNP or its buffer zone—the villages of Bodel, Mayo Yum, Gashaka, Selbe, Filinga, as well as the abandoned settlement of Yakuba. Stretching across a corridor of ca. 50 × 30 km, these 1,500 km$^2$ are colonized by a single baboon group, which inhabits the park's Southern Gashaka sector (*Higham et al., 2009*; *Ross et al., 2011*; *Warren et al., 2011*), where the vegetation is a mosaic of montane, submontane, lowland and riverine gallery forest with some proportions of Guinea woodland-savannah and grassland (*Adanu, Sommer & Fowler, 2011*). Troop sizes average 21 individuals, which typically include five adult females, three adult males, and offspring that are born throughout the year (the average number of males and subadults has not been reported) (*Higham et al., 2009*; *Jesus, 2019*). Most crania were found in the vicinity of human dwellings where they are more likely to be recovered but their locations suggest that all skulls stem from troops that engaged in crop-raiding to some extent. The single exception is one female skull, assumed to have been philopatric, which was found many kilometers away from the nearest cultivated fields and is not likely a crop-raider.

Each cranium was soaked for one day in a denture-cleansing powder solution to remove debris, given a field accession number, and stored in a plastic or metal box. As of April 2013, the collection comprised 111 skulls, including both adults and subadults. After excluding subadults, unprovenanced females, and specimens displaying extensive wear, damage, or a significant lack of antimeric teeth, an analytical sample of 81 individuals with antimeric tooth pairs remained. This study sample includes 43 adult males and 38 adult females.

## METHODS

### Data sharing

Data, scripts, and preliminary analysis outputs (*e.g.*, distribution testing, outlier tests, scatterplots) are available at GitHub: https://github.com/kchoover14/BaboonStress.

### Age and sex estimation

Age was assessed using the developmental stage of each tooth (*e.g.* unerupted, emerging or in occlusion) and the development of the basilar suture (*Kahumbu & Eley, 1991*; *Reed, 1973*). Adults are defined by full occlusion including the third molar—subadults demonstrate variable eruption but may have a full complement of permanent dentition in occlusion, with the exception of third molars. Sex was assessed visually in adults on the basis of overall skull size, robusticity, size of canines or canine orifice, and other known
dimorphic features that are easily observed (*Leigh & Cheverud, 1991*; *Singleton, 2002*; *Singleton et al., 2017*).

## Data collection

Standard maximum length and breadth measurements for bilophodont primates (*Swindler, 2002*) were taken 10 times (EG) on the permanent dentition of adult mandibular and maxillary premolar and molar teeth using Mitutoyo digital calipers—replicate measurements are used to assess the impact of measurement error (ME) on FA (see Choosing Replicates). The final dataset consisted of a maximum of nine variables per individual (we did not collect data on the maxillary third premolar due to canine honing). Because some individuals were missing teeth or exhibited traits that precluded measurement. (*e.g.*, cracked crowns, wear, poor preservation), data collection on all nine variables was not always possible for each individual.

## Choosing replicates

Because ME is a component of any metric value, an analysis of FA must consider the contribution of ME to FA. We had the luxury and burden of ten replicates for each trait—luxury because most studies take 2–3 replicates and burden because ten is an unwieldy number for analysis. Replicate measures were taken over a period of time with breaks between each data collection trial. Early trials of data collection may contain more ME due to a lack of familiarity with the teeth but later trials may contain more ME due to data collection fatigue. Thus, we assessed ME across sets of replicates (Table 1) to determine the minimum number of replicates with the lowest ME. ME, as a percentage of between-sides difference attributable to ME (*Palmer, 1994*), was calculated in the FA worksheet and uses the ME3 formula, which is a descriptor of ME independent of units of measurements (*Palmer, 1994*; *Palmer & Strobeck, 1986*; *Palmer & Strobeck, 2003a*). The full set of 10 replicates had the lowest mean ME (8%) but with a wide range (18%). These ME values are high for FA but can be reduced after excluding individual datum that fail the data inspection tests for outliers.

## Statistical methods—confounding factors affecting FA

Estimates of FA may be confounded by a variety of factors including bad raw measurements, high ME, aberrant individuals, directional asymmetry, antisymmetry, and trait size dependency (*Palmer & Strobeck, 2003a*). The data repository contains a step-by-step analysis of all factors affecting FA in this dataset (*Palmer & Strobeck, 2003b*) and is only briefly discussed here. Directional asymmetry may indicate biomechanical wear or side preference (such as handedness in humans) and is identified by having a significant skew (*Palmer & Strobeck, 1986*; *Palmer & Strobeck, 1992*). Antisymmetry may indicate a genetically controlled asymmetry (*e.g.*, one over-sized signalling claw in male fiddler crabs) that is randomly distributed between sides and is identified by having a platykurtic (broad-peaked) or bimodal distribution (*Palmer & Strobeck, 1986*). Directional asymmetry and antisymmetry are both assessed through skew and kurtosis statistics that describe

**Table 1 ME summary by replicate set.**

| Replicates | ME3 | Mean | Median | Minimum | Maximum | Range |
|---|---|---|---|---|---|---|
| 9–10 | 12% | 25% | 22% | 10% | 51% | 41% |
| 5–6 | 12% | 24% | 22% | 9% | 47% | 38% |
| 6–9 | 7% | 16% | 16% | 5% | 33% | 29% |
| 2–5 | 7% | 18% | 16% | 7% | 37% | 30% |
| 4–7 | 8% | 18% | 15% | 5% | 51% | 46% |
| 3–8 | 6% | 13% | 12% | 5% | 36% | 31% |
| 2–9 | 4% | 10% | 9% | 4% | 31% | 27% |
| 1–10 | 4% | 8% | 8% | 3% | 21% | 18% |

the distribution of the observed data (*Palmer & Strobeck, 1992*). Trait size variation also conflates measures of FA and is tested through the use of data visualization methods and statistical tests for outlier presence. Trait size dependency is a particular issue because FA may be different across traits within a species or the same trait across species (*Palmer & Strobeck, 2003a*) and can be tested by correlation between average trait size and average side difference.

DA was high in the sample. There is no obvious explanation but it may be caused by chewing side preference (*Martinez-Gomis et al., 2009*; *Nissan et al., 2004*). Some mammal species are known to have side-specific chewing preferences, such as horses (*Parés Casanova & Morros, 2014*) and goats (*Parés Casanova et al., 2018*). However, this has not been definitively observed in non-human primates, with the exception of some prosimians where side preference has been shown at the level of the individual (*Stafford, Milliken & Ward, 1993*). Interstitial wear impacts length measurements, so if the baboons in our study sample had a specific side preference, we would expect to see greater DA in length measurements compared to breadth, which is indeed what we see in our data. Irrespective of the mechanism underlying this, due to high DA, four variables had to be eliminated: female mandibular first and second molar lengths, and male mandibular first and second molar lengths.

Due to trait size dependency, one variable was eliminated (male maxillary third premolar length). Because ME was high and varied considerably across tooth, metric (*i.e.*, length, breadth), and sex, we used the FA10a index. The FA10a index is a measure of the magnitude of FA *after* parsing ME (Table 2). The total dataset for analysis consisted of nine variables for female breadths, 7 variables for female lengths, nine variables for male breadths, and six variables for male lengths. Table 2 displays the sample size per each variable's FA10a index to provide perspective on statistical power underlying the index.

## Statistical methods—data visualization
Data distributions for all ten replicates and plots of FA10a were visualized using ggplot2 (*Wickham, 2016*).

| Table 2 FA10a index values for hypothesis testing. | | | | |
|---|---|---|---|---|
| Sex | Metric | Tooth | FA10a | $n^1$ |
| Female | Breadth | MNM1 | 0.01 | 20 |
| Female | Breadth | MNM2 | 0.01 | 16 |
| Female | Breadth | MNM3 | 0.01 | 20 |
| Female | Breadth | MNP4 | 0.00 | 14 |
| Female | Breadth | MXM1 | 0.01 | 31 |
| Female | Breadth | MXM2 | 0.01 | 30 |
| Female | Breadth | MXM3 | 0.03 | 34 |
| Female | Breadth | MXP3 | 0.04 | 26 |
| Female | Breadth | MXP4 | 0.01 | 31 |
| Female | Length | MNM3 | 0.04 | 20 |
| Female | Length | MNP4 | 0.06 | 15 |
| Female | Length | MXM1 | 0.02 | 31 |
| Female | Length | MXM2 | 0.04 | 32 |
| Female | Length | MXM3 | 0.03 | 32 |
| Female | Length | MXP3 | 0.04 | 27 |
| Female | Length | MXP4 | 0.05 | 31 |
| Male | Breadth | MNM1 | 0.01 | 17 |
| Male | Breadth | MNM2 | 0.03 | 16 |
| Male | Breadth | MNM3 | 0.01 | 14 |
| Male | Breadth | MNP4 | 0.01 | 15 |
| Male | Breadth | MXM1 | 0.01 | 31 |
| Male | Breadth | MXM2 | 0.02 | 34 |
| Male | Breadth | MXM3 | 0.02 | 34 |
| Male | Breadth | MXP3 | 0.03 | 25 |
| Male | Breadth | MXP4 | 0.02 | 29 |
| Male | Length | MNM3 | 0.08 | 14 |
| Male | Length | MNP4 | 0.07 | 15 |
| Male | Length | MXM1 | 0.05 | 33 |
| Male | Length | MXM2 | 0.07 | 35 |
| Male | Length | MXM3 | 0.08 | 33 |
| Male | Length | MXP4 | 0.10 | 29 |

Note:
[1] The sample size from which the FA10 index was created.

## Statistical methods—hypothesis testing of research questions

Analysis was conducted by KCH in R v3.6.2 (*R Development Core Team, 2008*) using R Studio v.1.2.1335 (*RStudio Team, 2015*). Because FA is a measure of variance about the mean, Levene's test for equality of variance was used to test hypotheses in R via the leveneTest function in the car package (*Fox & Weisberg, 2011*). The eruption of upper and lower teeth varies only by a few months in baboon species, which allowed us to pool mandibular and maxillary molars for analysis (*Fortman, Hewett & Bennett, 2002*; *Hlusko & Mahaney, 2009*; *Phillips-Conroy & Jolly, 1988*; *Reed, 1973*).

**Table 3 FA10A trends.**

| Model | F-value | df | p-Value |
|-------|---------|-----|---------|
| FA10~Tooth | 1.78 | 8.00 | 0.14 |
| FA10~Class | 0.19 | 1.00 | 0.67 |
| FA10~Arcade | 0.26 | 1.00 | 0.61 |
| FA10~Metric | 8.48 | 1.00 | 0.01 |

**Table 4 Results of hypothesis testing (Levene's test).**

| Hypothesis | Result | Model | F | df | p |
|------------|--------|-------|-----|-----|-----|
| Sex differences | Significant sex differences | FA10~Sex | 10.602 | 1 | 0.003 |
| Weaning | No difference | FA10:M1~Tooth Type | 1.670 | 1 | 0.207 |
| | Significant sex differences | FA10:M1~Tooth Type*Sex | 6.696 | 3 | 0.002 |
| Reproduction | No difference | FA10:M3~Tooth Type | 0.064 | 1 | 0.802 |
| | Significant sex difference | FA10:M3~Tooth Type*Sex | 4.313 | 3 | 0.013 |

## RESULTS AND DISCUSSION

### Data exploration

In humans, FA varies by dimension (*e.g.*, length, breadth), arcade (*e.g.*, mandible, maxilla), and tooth class (*e.g.*, incisor, canine, premolar, molar) (*Bailit et al., 1970*; *Harris & Nweeia, 1980*). We explored the data to see if similar trends were apparent in our baboon sample (Table 3). We noted above that there are not currently enough data across primate species to understand if FA in non-human primates exhibits the same trends as humans. Our analysis is the first step towards understanding if human trends are visible in our data for olive baboons. There were significant differences in dimensions, with breaths exhibiting lower FA (0.02) and lengths exhibiting higher FA (0.06)—same as in humans (*Harris & Nweeia, 1980*). There were no significant differences by tooth class or arcade.

### Sex differences

The first research question is whether there is a sex-based difference in developmental instability. We tested the null hypothesis of no difference. Results indicate there are significant sex-differences in this sample (Table 4), suggesting that males experience greater developmental instability across the period of growth and development. Figure 1 shows the spread of FA10a index values for males and females. Both sexes exhibit greater FA in lengths compared to breadths and males have overall greater variance (FA is a measure of variance), even if some values overlap with females. Mandibular P4 exhibits the most FA in both males and females, but female FA values are almost half that of males.

### Instability during the weaning period

The second research question is whether weaning is a more stressful time compared to other developmental stages. Due to its developmental timing, the first molar is a likely
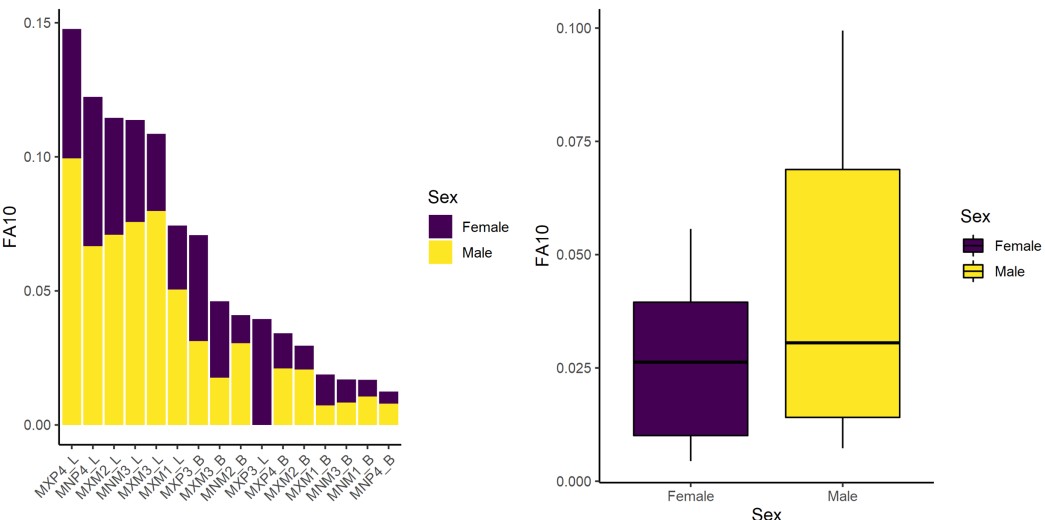

**Figure 1** FA10a values individually by sex (stacked plot, left) and grouped (boxplot by sex, right).

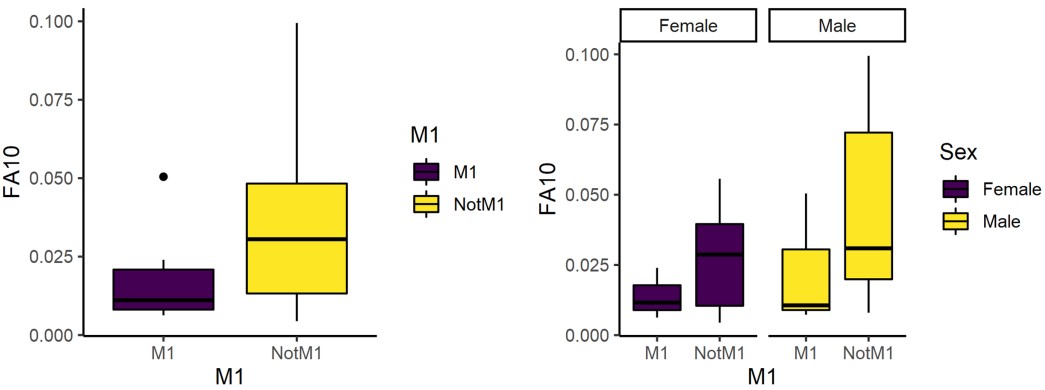

**Figure 2** FA10a comparing first molar to other teeth (left) and within sex (right).

diagnostic for weaning stress and was expected to have greater variance than the other teeth. FA10a is lower in M1 than other teeth, suggesting weaning is not more stressful compared to other stages of life history (Table 4). There are significant differences between male and female FA10a values, however (Table 4, Fig. 2). Both sexes exhibit greater FA in lengths compared to breadths and values for fourth premolars are again the highest (Fig. 3).

## Instability during the reproductive period

The third research question is whether reproduction is more stressful for females compared to males. We expected female third molars to have greater FA10a values than males and for FA10a to be higher in third molars compared to other teeth. FA10a is lower in M3 than other teeth when sexes are pooled, suggesting that weaning is not more stressful compared to other stages of life history (Table 4). There are significant differences between male and female FA10a values (Table 4, Fig. 2). While males have higher FA10a

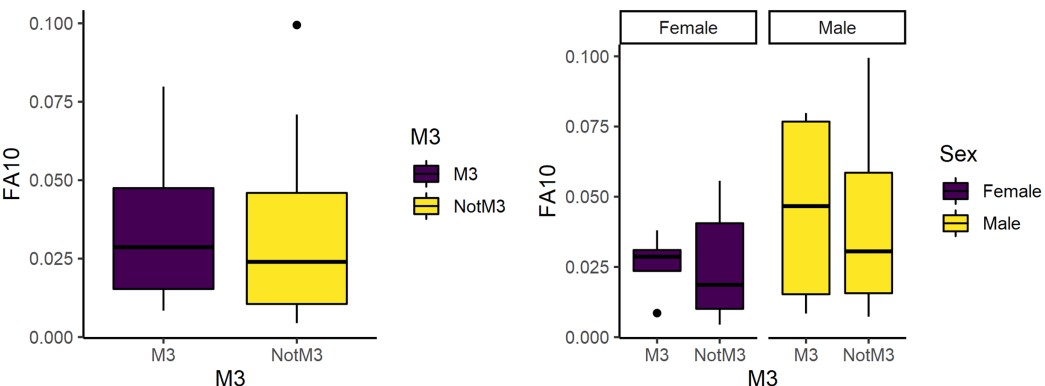

**Figure 3** FA10a comparing third molar to other teeth (left) and within sex (right).

values during reproductive years (based on the third molar developing during this life history stage), both sexes exhibit greater FA in lengths compared to breadths (Fig. 3).

## CONCLUSION

Due to the challenges in measurement presented by baboon teeth, we highly recommend researchers assess and reduce ME across multiple practice trials prior to data collection. We had a high number due to the novel nature of the study, but such a high number is not common nor recommended. High repeatability will minimize the number of replicates for analysis. A target number of replicates is typically 2-3 on 10% (or some other subset) of the study sample.

We used the FA10a index for the analysis of FA in our sample because it subtracts measurement error from the final index used for hypothesis testing. First, we examined FA across variables (length and breadth measures for individual teeth) to find any trends in FA. The only trend we found was that lengths exhibit more FA than breadths—a trend identified in humans as well (*Harris & Nweeia, 1980*). Second, we tested hypotheses about developmental instability in different life history phases (*i.e.*, weaning, reproduction).

### Sex differences

Males had greater overall FA values than females, suggesting greater levels of developmental instability. Male baboons have been observed to have relatively shorter and harder lives than females because they leave their birth groups and engage in more violent behavior (*e.g.*, chases, displays) when establishing and maintaining their place in the dominance hierarchy of the new group (*Cheney & Seyfarth, 2008*). Above and beyond the factors associated with general baboon social structure and hierarchy, males and females have differences in their developmental pathways too. The stability of male developmental pathways may be compromised by several factors. The male fetus stimulates additional maternal antigen production (*Lalumière, Harris & Rice, 1999*), which may cause male-biased prenatal mortality and other developmental complications (*Gualtieri & Hicks, 1985*). In addition, elevated testosterone profiles have been associated with immunosuppression (*Bradley, 1987*; *Folstad & Karter, 1992*; *Muehlenbein & Watts,*

_2010_; _Roberts et al., 2012_), the costs of which range from parasite susceptibility to developmental instability. Given these ontogenetic challenges, males may be more susceptible to external environmental conditions, such as the extremely wet climate and high pathogen exposure during the rainy season that baboons experience at GGNP (_Higham et al., 2009_). Other researchers have observed increased FA in marginal or less favorable environments (_Bailit et al., 1970_; _Parsons, 1992_) and males may be at a disadvantage in these circumstances.

## Weaning

FA values were lower in first molars than other teeth, suggesting that weaning is not more stressful compared to other life history stages. Compared to males, however, females had greater overall FA values for the first molar, suggesting a comparatively more stressful time than males during weaning. Mother-infant contact reduces steadily with growth. Mothers reject suckling attempts as early as six months of age but most vigorously when the infant approaches a year (_Nash, 1978_; _Rhine et al., 1985_). This schedule corresponds with first molar development. For yearling baboons, developmental instability may be linked to weaning stressors which include (a) nutritional stress from decreasing energy availability (_Altmann, 1998_), (b) psychosocial stress from increased separation from the mother (_Levine, 2005_) and (c) physiological stress in the form of decreased pathogen resistance (_Katzenberg, Herring & Saunders, 1996_).

## Reproduction

FA values were lower in third molars than other teeth, suggesting that reproduction is not comparatively more stressful than other life history stages. Males, however, had higher third molar FA values than females, suggesting they have a comparatively more stressful time during reproductive years then females. In olive baboons—and a wide range of other primate species—female philopatry and male dispersal is the rule. As mentioned above, baboon males have generally shorter and tougher lives (_Cheney & Seyfarth, 2008_). There are multiple factors, however, that may compromise the fitness of emigrating individuals. A lone male is more vulnerable to predation (_Dunbar, 1987_) and, as such, spends less time foraging (_Slatkin & Hausfater, 1976_). Elongated solitary periods of recently matured males also impede their mating prospects. Organisms under such dietary and reproductive stresses expend more energy to counter these challenges (_Parsons, 1990_). The stress associated with male dispersal, both nutritional and psychosocial, may be greater than that of females despite the clear biological burdens of reproduction on the female body. The third molar is the last tooth to develop and has been noted to exhibit greater morphological variation perhaps due to relaxed selective pressure—this might make it more susceptible to the ontogenetic effects of sexual dimorphism (_Butler, 1939_; _Gingerich, 1974_; _Mayhall & Saunders, 1986_). Third molar length exhibits greater variation than other teeth (_Gingerich & Schoeninger, 1979_) and the hypoconulid on the distal surface of cercopithecoid mandibular third molars (_Swindler, 2002_) may act to increase variance and asymmetry. Finally, given the negative association (in gorillas) of crown height and FA,

we might expect that lower ranked males, who capture a larger share of any sample, would have higher FA from developmental instability, particularly when secondary sexual traits are developing (*Manning & Chamberlain, 1994*).

Females in our baboon sample have previously been found to have elevated glucocorticoid levels that measure physiological stress (*MacLarnon et al., 2015*). Our data suggest females exhibit greater instability than males only during weaning. Further, when comparing our data for males to females across all variables, we find that males exhibit greater developmental instability during growth and development than females and developmental instability is highest during early reproductive years. Taking a broader view across life history, we note that our data suggest that reproduction (using M3 FA10 as a proxy) is much more stressful than weaning (using M1 FA10 as a proxy) for both males and females, with mean FA values almost double that for reproduction and almost triple when considering males alone (Figs. 2 and 3). Given that male baboons lead relatively harder and tougher lives than females, our findings are in line with the larger understanding of the effect of social structure on health (*Cheney & Seyfarth, 2008*). Demonstration of FA and the supposition that a group appears under developmental instability can be relatively straightforward (*Leary & Allendorf, 1989*), but the identification of a specific stressor remains conjectural. Habitat quality (*Manning & Chamberlain, 1994*), psychosocial factors (*Newell-Morris, Fahrenbruch & Sackett, 1989*), diet (*Swaddle & Witter, 1994*) and social hierarchy (*Cheney & Seyfarth, 2008*) have all been implicated.

## ACKNOWLEDGEMENTS

We thank the following individuals for contributing to our ideas at various stages of this project: Emily Creaser, Wendy Dirks, Sarah Elton, Chris Gilbert, Kieran McNulty, Richard Palmer, Christophe Soligo and the PalaeoMorpho Research Group at Durham University. KK and KCH also thank WWB.

### Funding

This work was supported by the Durham University Learning and Teaching Award and the North of England Zoological Society/Chester Zoo. The funders had no role in study design, data collection and analysis, decision to publish, or preparation of the manuscript.

### Grant Disclosures

The following grant information was disclosed by the authors:
Durham University Learning and Teaching Award.
North of England Zoological Society/Chester Zoo.

### Competing Interests

Kara C Hoover is an Academic Editor for PeerJ.

## Author Contributions

- Kara C. Hoover performed the experiments, analyzed the data, prepared figures and/or tables, authored or reviewed drafts of the paper, and approved the final draft.
- Emily Gelipter conceived and designed the experiments, performed the experiments, authored or reviewed drafts of the paper, and approved the final draft.
- Volker Sommer conceived and designed the experiments, authored or reviewed drafts of the paper, and approved the final draft.
- Kris Kovarovic conceived and designed the experiments, performed the experiments, authored or reviewed drafts of the paper, and approved the final draft.

## Field Study Permissions

The following information was supplied relating to field study approvals (*i.e.*, approving body and any reference numbers):

The Nigeria National Park Service approved research permits to undertake work at Gashaka Gumti National Park NPH/GEN/378/V/504.

## Data Availability

The data and scripts are available at GitHub: https://github.com/kchoover14/BaboonStress.

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
