# Peer review of "Developmental instability in wild Nigerian olive baboons (Papio anubis)"

_PeerJ, doi:10.7717/peerj.11832_

## Round 0.1 · original submission · Minor Revisions

I apologize for the delay in getting this decision to you. The previous editor has been out of touch and I have been asked to follow up. The three reviewers have much more expertise than I do on this topic so I will defer to their opinions. Thankfully, they were all similarly positively disposed to your MS, requesting only minor revision. I have read the MS myself and agree with their assessment that it is clear and well-written. This is a wonderfully well-referenced yet concise paper with a clear rationale.

On line 88, perhaps you could insert "most" before "other mammals" as there are other large-bodied mammals, and also smaller-bodied mammals, such as chiroptera with slow life histories.

On line 287, and elsewhere, do you really mean "reproduction" here? Perhaps it would be better to write "the reproductive period"

·

Basic reporting

The article is written in a clear and direct manner that avoids the overuse of jargon. A concise overview of fluctuating asymmetry is provided, along with a brief discussion of factors that may confound such an analysis. Appropriate references are made to previous research on the subject (specifically, Palmer and Strobeck, Bailit, Garn, Perzigian, etc...). The structure and organization of the article is good; I found little reason to backtrack, as each section builds in an appropriate way on the last. Figures and tables are labeled appropriately and convey the results of the research well.

There were, however, several things that I felt might have improved the communication of this research to an audience. It would have been helpful for the authors to include the actual ages (in years or months) at which weaning and sexual maturity occur in relation to when crown development is completed, as a means of demonstrating that the timing of these stressors is correlated to crown asymmetry. It also might have been useful to provide additional comments within the R scripts in the data repository or file names that communicate the purpose/function of these files -- the data and the scripts themselves, however, were excellent. It might also have been beneficial to include a table of the ANOVA results.

There is a typo on line 82 -- I believe this is meant to read "is non-directional" rather than "in non-directional"

Experimental design

The research, which tests three interrelated hypotheses regarding fluctuating asymmetry (FA) in baboons, certainly fits within the aims and scopes of the journal. The research questions are well defined and the purpose of the research itself (as FA has been largely ignored in non-human primates) is certainly relevant.

The experimental design follows the recommendations of Palmer and Strobeck (1986) and Palmer (1994) for an analysis of fluctuating asymmetry. Measurement error was estimated for replicate data, outliers were removed when appropriate, and the resulting data sets were tested for deviations from normality and confounding factors such as directional asymmetry and antisymmetry. Estimation of FA, utilizing the FA10a index value which removes measurement error as part of a two-way mixed model ANOVA, resulted in valid estimates of FA for the groups under study.

I felt that two aspects of the methods might have been discussed in more detail. The actual process of determining whether or not these traits were size dependent (and why this is important) might have been discussed in more detail. Likewise, the method for detecting antisymmetry through an analysis of skewness and kurtosis could also be outlined in more detail. The article, on line 233-235 suggests that these methods are outlined in the data repository -- rather than go through an external source, I would have liked to see this discussion in the article itself.

Validity of the findings

The test results for the three hypotheses are clear and direct; the authors demonstrate through their analysis that estimates of FA differences between males and females, and differences by sex with regards to weaning and sexual maturity, were significant. The authors have, generally, avoided unnecessary speculation about the cause and effect relationship between FA and specific stressors.

I do question, however, the reference to Bailit (1970) on line 330. The authors suggest that human studies of FA are inappropriate for comparison to non-human primates (lines 105-111) yet they are comparing their own results to those of Bailit which, in addition to being based on a human sample, are based upon a less rigorous analysis of FA.

Additional comments

A well designed and executed analysis of fluctuating asymmetry within Papio anubis that makes a long overdue and meaningful contribution to the scholarship on developmental stress in non-human primates.

·

Basic reporting

no comment

Experimental design

no comment

Validity of the findings

no comment

Additional comments

This is a very nice contribution to the science of non-human primate dental studies, with relevance to mammalian growth and development more generally. The authors have written a clean manuscript that makes their study design clear, their results easy to see, and their conclusions in line with the scope of the strengths of the study design and their results. The observation that FA varies differentially between males and females is very interesting. This window into the stress differences that males and females experience during their maturation is intriguing.

I have just three recommendations:
1) Add a sentence to the abstract noting the environmental “extreme-ness” of this population among baboons.
2) Add in a few sentences answering the question of how interstitial wear may be playing into the FA magnitude difference observed between mesiodistal and buccolingual measurements.
3) Make the colors and labeling of the three bar charts more user-friendly.

Reviewer 3 ·

Basic reporting

The language use is standard and clear. I have noted a few minor exceptions with comments in the marked up pdf of the manuscript.

The literature is appropriate, though I feel the authors missed an opportunity to strengthen their argument for the use of FA as a stress indicator, by not citing any of the literature that addresses problems with identification and error associated with the analysis of dental defects.

The structure, tables, and figures are appropriate, and the authors state they are sharing their raw data.

The hypotheses are testable and tested. I suggest the rewording of one hypothesis, and the possible re-consideration of a second. This reconsideration should not be a requirement of publication.

Experimental design

We know almost nothing about NHP dental FA, so this paper definitely fills a gap.

The investigation is rigorous, although perhaps more than need be. I don't understand why the authors took 10 sets of measures. An explanation for this choice should be made, especially as they point out that it made their analysis more complicated.

The methods are duplicable as described.

Validity of the findings

The only issue here is the overstatement of the purity of results. FA is a result of interactions among environmental stressors and somatic buffering/frailty. The authors do not weigh their results interpretations sufficiently because they do not address the unknown effects of buffering. Please see specific comments made in the MS pdf.

Otherwise, all is reasonable.

Additional comments

Thank you for writing a straightforward, easy to read paper, that makes a very specific contribution. If you address the comments made here and in the pdf, the paper should be publishable in short order.

Annotated reviews are not available for download in order to protect the identity of reviewers who chose to remain anonymous.

---

## Round 0.2 · accepted · Accept

Because the previous reviewers all recommended only minor revisions, and because your response and revision appear to be detailed, and highly responsive to the reviewers' suggestions, I have elected not to send your revision back out for re-review. I was fortunate that the reviewers were very clear and thorough in their initial reviews. I appreciate your openness to the reviewers' comments and suggestions and your care in considering alternative interpretations of the data. I am therefore pleased to accept your paper. However, I did notice the following typos that should be corrected during proofing. Line numbers refer to the tracked changes word file.

Line 32. I believe "is" should be deleted.

Line 45. I find it odd to refer to reproduction as a life history stage, especially for males. Can this be clarified before this statement?
Be consistent in using nonhuman rather than non-human throughout.

Line 143 is missing an "in" between sex differences and developmental instability.

Line 550 place a , after formula.

Line 632 "on" is not needed after "impacts."